# Examining Predictors of Different ABA Treatments: A Systematic Review

**DOI:** 10.3390/bs12080267

**Published:** 2022-08-04

**Authors:** Mariangela Cerasuolo, Roberta Simeoli, Raffaele Nappo, Maria Gallucci, Luigi Iovino, Alessandro Frolli, Angelo Rega

**Affiliations:** 1Associazione Italiana Per L’Assistenza Spastici Onlus Sez Di Cicciano, 80033 Cicciano, Italy; 2Faculty of Medicine, University of Ostrava, 70103 Ostrava, Czech Republic; 3Department of Humanistic Studies, University of Naples Federico II, 80138 Napoli, Italy; 4Neapolisanit S.R.L. Rehabilitation Center, 80044 Ottaviano, Italy; 5Disability Research Centre, Department of Sociology and Social Research, University of International Studies of Rome, 00147 Rome, Italy

**Keywords:** predictor, applied behavior analysis, autism, treatment outcome

## Abstract

In the recent literature, there is a broad consensus on the effectiveness of Applied Behavior Analysis interventions for autism spectrum disorder (ASD). Despite their proven efficacy, research in clinical settings shows that these treatments are not equally effective for all children and the issue of which intervention should be chosen for an individual remains a common dilemma. The current work systematically reviewed studies on predictors and moderators of response to different types of evidence-based treatment for children with ASD. Specifically, our goal was to critically review the relationships between pre-treatment child characteristics and specific treatment outcomes, covering different aspects of functioning (i.e., social, communicative, adaptive, cognitive, motor, global functioning, play, and symptom severity). Our results questioned the binomial “better functioning-better outcome”, emphasizing the complex interplay between pre-treatment child characteristics and treatment outcomes. However, some pre-treatment variables seem to act as prerequisites for a specific treatment, and the issue of “what works for whom and why” remains challenging. Future research should focus on the definition of evidence-based decision-making models that capture those individual factors through which a specific intervention will exert its effects.

## 1. Introduction

The fifth edition Text Revised of the Diagnostic Statistical Manual (DSM-5-TR [1]) and the latest edition of the International Classification of Disease (ICD-11) concur in defining the autism spectrum disorder (ASD) as a neurodevelopmental disorder characterized by persistent and pervasive social–emotional reciprocity and social communication impairments, and by the presence of restricted and repetitive behaviors, interests, and activities. Apart from these core symptoms, another feature of ASD is the high clinical variability, which hampers the diagnostic process and the selection of the most suitable treatment. 

Currently, several behavioral models have been developed and have proved to be effective in improving communication, socialization, adaptive, and cognitive functioning in autism [2]. Specifically, evidence-based treatments for ASD are based on the principles of Applied Behavior Analysis (ABA) and are usually grouped into two approaches: Early Intensive Behavioral Intervention (EIBI) and Naturalistic Developmental Behavioral Intervention (NDBI) [3].

EIBI is a comprehensive intervention for very young children with ASD, usually implemented intensively in an individualized format. EIBI exploits principles of ABA to teach specific discrete skills generally in highly structured sections known as Discrete Trial Training (DTT [4]). NDBIs [5] share with EIBI models the focus on early ages, the intensive delivery format, and the use of evidence-based practice and manualized procedures [6]. NDBIs merge ABA principles with strategies from developmental psychology, implementing them in naturalistic and interactive social contexts. Unlike EIBI models, NDBIs promote the acquisition of language, social, and play behaviors through child-directed teaching strategies and natural reinforcement contingencies in social–communication–play interactive sections, following the child’s intrinsic motivation [5]. Examples of NDBIs include comprehensive interventions, such as Incidental Teaching (IT [7,8]), Pivotal Response Training (PRT [9,10]), the Early Start Denver Model (ESDM [11,12]), and focused interventions such as Reciprocal Imitation Training (RIT, [13,14]), Joint Attention Symbolic Play Engagement and Regulation (JASPER, [15,16]).

Despite their proven efficacy, research in clinical settings shows that these treatments are not equally effective for all children, highlighting a high heterogeneity at an individual level in response to intervention (for reviews, see [17,18,19]).

To better understand outcome variability, several studies have been conducted with the aim of identifying the pre-treatment variables that may influence treatment response. Studies identifying predictors of treatment response are useful to guide practitioners in the decision-making process for treatment selection and adaptation. According to Yoder and Compton [20], two research approaches have been used to detect predictors of treatment response. The first consists of correlational or regression analysis between variables at intake and changes in dependent variables after a definite treatment. The second approach uses a series of single-subject experiments to identify treatment responder and nonresponder’s profiles and to compare them to detect possible differences prior to the intervention.

Although literature in this field is scarce, it has been frequently pointed out that having pre-treatment abilities in cognitive, social, and language functioning would predict better treatment response [21,22,23]. However, it is likely that the issue is more complex than the simple binomial “better functioning-better outcome”. As highlighted in a recent paper by Chen and coworkers [19], this line of research is limited in elucidating what works for whom, why, and when, given the dynamic nature of the disorder and that of the treatment.

To predict whether and to what extent a certain child could respond to a definite intervention, practitioners should focus on both the strengths and weaknesses of each child and how these individual characteristics intertwine with those of the treatment.

Here, we aim to shed light on this issue by systematically reviewing the literature on child characteristics that may influence response to specific treatment approaches. Furthermore, we will discuss implications in clinical practice and, specifically in treatment selection, based on evidence collected.

## 2. Methods: Data Sources and Study Selection

Extensive research was carried out by two properly trained reviewers (M.C. and R.S.) on Pubmed and by analyzing the reference list in Vivanti et al. [17] to locate studies addressing predictors and moderators of target treatment response. The search strategy consisted of the following keywords: (autism) AND (predictor) AND (response OR outcome OR efficacy OR effectiveness)) AND (intervention OR treatment). The parentheses group terms together. In the above search, at least one term from each of the four groups had to be present in the title or abstract.

Articles were included if:(a)the sample included individuals with a clinical diagnosis of autism, infantile autism, ASD, PDD-NOS, or Asperger’s Syndrome (AS) in childhood (with and without an intellectual disability) and were younger than 13 years old(b)an ABA-based comprehensive behavioral treatment was administered(c)predictors and/or moderators were considered and referred to the patient’s individual characteristics(d)written in English.

No setting (e.g., home, school/kindergarten/other education settings, clinic) or study design limits were imposed. Systematic reviews and meta-analyses were excluded, together with study protocol papers.

Studies were excluded if participants were adolescents (older than 13 years old); treatment was not clearly specified or did not meet our inclusion criteria; no predictors of treatment responses were considered; predictive variables concerned parental, intervention, or therapist characteristics; interventions were parent-mediated.

The titles and abstracts of the search results were then screened, and the relevant papers were identified.

In addition, the earlier review by Vivanti et al. [17] was examined, and relevant studies not identified in the search described above were included if they met our inclusion criteria.

As shown in Figure 1, database searches produced 533 records, and 10 additional articles were selected from the reference list of a seminal paper by Vivanti et al. [17]. After the screening of titles and abstracts, 85 full-text articles were obtained for further analysis, and 21 articles met our inclusion criteria. Selected articles were primarily divided based on the method of identifying predictors of response treatment (whether correlational/regression analysis or responders profile analysis), then organized according to the class of predictors assessed (i.e., demographics, symptom severity, play, and social skills, cognitive functioning, communication, adaptive and maladaptive behaviors, developmental quotient, and emotional and psychiatric difficulties).

## 3. Results

Our review has been organized by separately describing all the moderating variables that have shown to affect treatment outcomes, arranged for behavioral macro-categories. Specifically, subsequent paragraphs provide an overview of different moderators affecting treatment response, according to the selected studies. The last paragraph summarizes all resulting studies focusing on responders’ profiles to specific behavioral interventions.

All included studies are summarized in Table 1.

### 3.1. Demographics

Age has been frequently studied as a predictor in several ESDM studies. For instance, Vivanti and coworkers [26] investigated whether treatment outcomes differed according to age by splitting the sample into two groups: autistic children aged 18–48 months and autistic children aged 48–62 months. The authors found that younger children achieved superior verbal developmental quotient (DQ) gains compared to older ones. No group differences emerged for nonverbal DQ and adaptive behavior or symptom severity. In addition, the authors found that the association between age and the verbal delay was moderated by baseline verbal level, indicating that younger children may benefit more from the ESDM intervention when they start with lower verbal skills [26]. In a previous study by the same research group [25], chronological age was negatively associated with expressive language but was not related to receptive language, visual reception, and fine motor as measured by the Mullen Scales for Early Learning (MSEL [47]), and to symptom severity based on Autism Diagnostic Observation Schedule Second Edition (ADOS-II [48]) scores. In contrast, two NDBI studies found no association between age at intake and verbal communication gains after 1-year of PRT intervention [31] and between age and ESDM domains changes after 1 year of ESDM treatment [30].

Only two correlational studies explored the predictive role of gender on treatment outcomes. In an interventional parallel group after an EIBI program [45], no associations were found between gender and adaptive and cognitive functioning. Contrasting results emerged from a more recent longitudinal study [33], showing that male participants tended to improve more quickly in adaptive functioning and physical development after an EIBI program. Instead, gender was not associated with social–emotional and communication domains as measured by the Developmental Profile-3 (DP-3 [49]) [33].

### 3.2. Symptom Severity

Symptom severity is classically measured through the ADOS-II score, addressing two behavioral domains: the Social Affect (SA) and Restricted and Repetitive Behaviors (RRB) domains.

Symptom severity has been frequently considered by studies exploring predictors of treatment response, reporting inconsistent results. While in a study by Vivanti and coworkers [25], symptom severity was found to be negatively associated with both expressive and receptive language changes after ESDM treatment, other studies did not find any relationship between autistic symptoms and language and communication skills after ESDM [30], EIBI [33,39]) and PRT [31] interventions. Similar inconsistent results emerged for IQ changes, with one EIBI study [39] showing that symptoms severity predicted cognitive changes, while others indicated no significant relationship between the two variables [25,42].

Two studies reported that more severe autistic symptoms were associated with a decrease in post-treatment socialization scores [30,39]. Instead, in Tiura et al. [33], no associations emerged between symptom severity and the social–emotional domain changes, together with those in the adaptive functioning after an EIBI intervention. Symptom severity at intake resulted in being a predictor of motor domain gains after EIBI [33] and ESDM treatments [30]. Instead, in a study by Vivanti et al. [25], baseline symptom severity was not associated either with motor changes or with improvements in autistic symptomatology. Finally, Magiati et al. [34] found that symptom severity was not predictive of treatment outcomes after an EIBI treatment.

### 3.3. Play Skills

According to our results, here we considered play skills as the functional use of objects and actions with objects. Notably, the role of these skills in modulating treatment outcomes has been investigated only in NDBI studies. Functional use of objects was found to be positively associated with visual reception, fine motor, and expressive language changes but not with receptive language and symptom severity after a group-based ESDM intervention [25]. Similar results emerged in a study by Fossum and coworkers [31], where toy contact was positively associated with expressive language gains after a PRT intervention.

Finally, in Contaldo et al.’s work [30], actions with objects, as measured by The MacArthur-Bates Communicative Development Inventories (MB-CDI [50]), were positively associated with socialization, cognition and play, and motor domains and with the number of learning objectives acquired by each child in a month, but not with communication domains, measured through the ESDM Curriculum Checklist.

### 3.4. Social Skills

The social skill domain includes a variety of abilities through which individuals interact with their social environment. Here, we considered in this category the following variables: social skills, attention to face or social attention, avoidance/social approach, joint attention, imitation, goal understanding, and socially-mediated reinforcers.

According to our results, general social skills have only been considered in EIBI studies and were mainly measured through the Vineland Adaptive Behavior Scale VABS-II [51]. Specifically, in Sallow and Graupner’s study [39], social skills were found to predict IQ changes, whereas the model was not significant for post-treatment language gains. Instead, no association was found between baseline social skills and IQ changes after a 3-year STAR program [42]. Smith and coworkers [40] have studied the role of the social engagement domain at intake, which includes the social approach, joint attention, and imitation, and found that higher social skills predicted higher cognitive and adaptive functioning over time but not changes in autism severity.

The ability to imitate others was positively associated with improvements in several aspects of functioning after ESDM-based intervention, such as visual reception, fine motor, and receptive language, but not with expressive language and symptom severity [25]. Instead, in another ESDM study [30], imitation was not associated with improvements in any domain of the ESDM Curriculum Checklist nor with the rate of learning objectives acquisition. Sallows and Graupner [39] found that pre-treatment ability to imitate predicted social skills, IQ changes, and language acquisition.

In two ESDM studies, the ability to direct attention to others and/or social stimuli resulted in affecting post-treatment improvements. Specifically, Robain et al. [29] found that children classified as social responders at intake reported decreased symptom severity, driven by a significant decrease in the social affect domain scores and higher developmental quotient compared to geometrical responders. Similarly, children with more attention to face at baseline reported higher developmental quotient over time, especially in the verbal domain [28]. In contrast, Vivanti et al. [25] did not find any relationship between social attention, measured through an eye-tracking paradigm, and MSEL subscale scores (visual reception, fine motor, expressive and receptive language) and symptom severity change over time. Moreover, the same authors found that goal understanding, defined as the ability to attribute meaning and purpose to others’ actions, was positively associated with receptive language and explained alone 30% of the variance. Instead, no associations were found between this variable and expressive language, visual reception, fine motor domains, and symptom severity improvements over time [25].

Fossum and coworkers [31], in their prospective single-subject design, found that lower social avoidance predicted PRT treatment outcomes in expressive language.

Finally, Klintwall and Eikeseth [41] focus on the type and number of reinforcers and found that the presence of several socially-mediated reinforcers at intake predicted an increase in the learning rate of adaptive abilities.

### 3.5. Cognitive Functioning

Several studies considered Cognitive variables as predictors of treatment outcome. All the studies considered in this work provided an IQ measure obtained through standardized tests such as: Griffiths’ Developmental Scales (GMDS [52], Merrill-Palmer Revised Scales of Development (MPR [53]), Psychoeducational Profile Third edition (PEP-3 [54]) and Wechsler Preschool and Primary Scale of Intelligence (WPPSI [55]), the DP-3, the Bayley Scales of Infant Development (BSID [56]), and MSEL.

According to our results, three ESDM studies and eight EIBI studies considered IQ as a predictor of treatment outcome.

Most of these studies analyzed the effect of IQ on adaptive behavior [33,34,35,40,43]. Ben-Itzchak et al. [43] observed the effect of IQ on all the VABS domains except for motor skills and receptive language subdomains. Tiura et al. [33] showed that children with higher cognitive functioning had rapid growth in communication, social–emotional, physical, and adaptive skills. Magiati et al. [34] also noted that IQ levels were predictive of cognitive skills, language skills, adaptive behavior, and autism severity. The research groups of Smith [38] and Eldevik [45] highlighted the significant effects of IQ on the Communication domain. In Sallows and Graupner’s study [39], the effects of IQ on social skills and language were not confirmed. They measured language through the Reynell Developmental Language Scales (RDLS [57]) and social skills using VABS.

The prediction effect of IQ was also analyzed on symptom severity reduction. Ben-Itzchak et al. [43] noted that IQ does not predict changes in symptom severity after treatment, contrasting with Smith et al.’s works [38,40] which reported an effect of IQ on symptom severity decreases after 1-year of EIBI treatment.

In addition, Smith et al. [38] also investigated the effect of IQ on problem behavior, measured through the MPR, revealing that IQ measure at intake did not predict the decrease in problem behavior after treatment [38].

### 3.6. Communication

In this study, we considered the communication domain as: expressive and receptive language, social–communicative gestures, lexical comprehension, and word production. The authors assessed these subdomains through different standardized tests, such as the Pervasive Developmental Disorders Behavior Inventory (PDDBI [58]), the Adaptive Behavior Assessment System Second edition (ABAS-II [59]), the British Picture Vocabulary Scales—Second edition (BPVS [60]), the Expressive One-Word Picture Vocabulary Test (EOWPVT [61]), the Communication subdomain of the VABS-II.

Contaldo et al. [30] analyzed the number of communicative gestures at baseline and observed a positive effect on communication, socialization, cognition, play, and motor domains after an ESDM treatment. They also found a positive association between Receptive Lexical Comprehension at baseline and socialization, cognition and play, and motor domains. In contrast, they did not find any significant association between Word Production and the aforementioned variables [30].

Results from Laister et al.’s study [36] suggest that parental rating of social use of gestures, assessed through the PDDBI, was the single strongest predictor of nonverbal DQ after ESDM intervention and a strong predictor of verbal gains.

Sallows and Graupner [39] conducted a study considering the effect of communication on EIBI outcomes. They considered communication as a measure of receptive language, showing an association with increased IQ level after treatment and language acquisition. The authors did not find any significant association between communication and social improvements after treatment [39].

Pellecchia et al. [42] also investigated the predictive effect of communication on IQ gains, but they did not confirm previous results. In their study, language and communication, measured with PDDBI and ABAS-II, did not predict any change in IQ measured with the Differential Ability Scales, Second edition (DAS-II [62]) [42].

Magiati and coworkers [34] showed that language skills assessed through BPVS and EOWPVT were predictive of EIBI outcomes after 4–5 years of intervention. They measured outcome gains as cognitive skills, language skills, adaptive behavior, and symptom severity measured with the Autism Diagnostic Interview-Revised (ADI-R [63]).

### 3.7. Adaptive Behavior

According to our results, four EIBI studies evaluated the predictive effect of “adaptive behavior” on treatment outcomes.

Magiati et al. [34] highlighted a positive effect of adaptive behavior at intake on cognitive, communication, adaptive behaviors, and autism severity. Sallows and Graupner [39] analyzed the effect of each VABS subscale and their association with post-treatment IQ, language, and social skills. As for daily living skills, the authors showed the predictive role of language domain improvements after treatment.

However, studies by Pellecchia et al. [42] and Klintwall and Eikeseth [41] did not confirm these results. In particular, Pellecchia et al. [42] showed that adaptive behavior at intake was not significantly associated with IQ improvement after treatment. Finally, Klintwall and Eikeseth [41] did not find any significant association between VABS score at intake and VABS learning rate during and after treatment.

### 3.8. Maladaptive Behavior

In this study, we considered maladaptive behavior the verbal self-stimulatory behavior, restricted interest and repetitive behaviors, problem behavior, and sensorimotor rituals measured through the Repetitive Behavior Scale-Revised (RBS-R [64]).

Two studies considered maladaptive behaviors as isolated variables with the potential to predict specific treatment outcomes [29,40].

Robain et al. [29] showed that higher maladaptive behavior was associated with lower DQ after ESDM treatment. Smith et al. [40] analyzed the effect of sensorimotor rituals on EIBI outcome and did not find any prediction effect after 1 and 2 years of intervention on IQ, adaptive behavior, social skills, and symptom severity.

### 3.9. Developmental Quotient (DQ)

In this study, we considered DQ as the global development derived from the composite score of tests such as the MSEL or GMDS.

According to our results, Contaldo and her colleagues [30] highlighted a significant effect of DQ on treatment outcomes. They showed that DQ at baseline was positively associated with gains in the ESDM socialization, cognition and play domains, and with the number of learning objectives acquired by each child in a month [30].

Vivanti et al. [25] did not confirm these results revealing that DQ at intake was not associated with visual reception, fine motor, receptive and expressive language, and symptoms severity after an ESDM treatment.

### 3.10. Emotional and Psychiatric Difficulties

We considered this variable as consisting of specific emotional aspects such as positive affect and symptoms associated with other psychiatric disorders such as attention deficit hyperactivity disorder (ADHD), social phobia, depression, anxiety, and conduct problem.

Pellecchia et al. [42] considered symptoms associated with common co-occurring psychiatric difficulties measured through the Child Symptom Inventory-4 (CSI-4 [65]) on IQ improvement. Results indicated that social anxiety symptoms significantly predicted student outcome IQ (DAS-II). In fact, each point increase on the CSI-4 scale for social phobia was associated with a decrease in the DAS-II score. Fossum et al. [31] analyzed the effect of positive affect on PRT outcomes revealing a significant association with expressive language after treatment.

### 3.11. Responders

Here, we reported the resulting studies exploring predictors of treatment response considering pre-treatment individual differences between high responders and low responders to a definite intervention.

Two studies explored “responder” profiles of ESDM treatment [27,36]. In Sinai-Gavrilov et al.’s work [27], preschool-based ESDM (PB-ESDM) treatment high responders were characterized by lower symptom severity and higher DQ, as observed by significant between-group differences in each MSEL subscale (i.e., visual reception, fine motor, expressive and receptive language) and higher adaptive behavior at intake compared to PB-ESDM treatment low responders. Instead, no between-group differences emerged for gender and age.

Laister et al.’s study [36] highlighted baseline differences between children with low and high language gains in response to ESDM intervention. According to their results, there were no differences between the two groups concerning age. Interestingly, no significant group differences were found for the parent-reported social approach behavior domain. Symptoms severity at baseline did not significantly differ between the two subsamples. Furthermore, receptive and expressive language and problem behavior were not significantly different for the two groups. Analyzing baseline differences between the two groups emerged that verbal and nonverbal DQ, fine motor skills, visual reception, and gestural approach behavior were significantly higher in the high responder group who gained the most in social skills and language skills as ESDM outcome [36].

As for PRT treatment, the research group of Sherer and Schreibman has focused on the behavioral profile characterizing PRT responders. In their earlier seminal study, Sherer and Schreibman [37] found that children who appropriately engaged more with toys avoided people less and produced more stereotyped and repetitive vocalizations or verbalizations at baseline responded better to the PRT treatment than nonresponders. Later, the same authors [32] studied those children whose behavioral profile matched the Sherer and Schreibman [37] “nonresponder” profile except for one behavior: either toy contact or avoidance. Results showed that children with high object interest might benefit more from a PRT treatment, although to a moderate extent, whereas those with low social avoidance remained in the nonresponder group [32]. A recent study [31] expanded Sherer and Schreibman’s results [32,37] and stated that high responders had greater toy contact, lower level of social avoidance, and stereotyped and repetitive vocalizations at baseline compared to low responders. Furthermore, no between-group differences were found for chronological age, symptoms severity, expressive language, cognitive ability, and positive affect. More specifically, the authors claimed that levels of expressive language, cognitive ability, toy contact, positive affect, social avoidance, and stereotyped and repetitive vocalizations predict treatment outcome in a directly proportional and linear manner, in that higher levels predict a stronger response [31].

Three studies explored differences between high and low responders of the EIBI program. In Remington et al. [44], children who responded most positively to the intervention had higher IQ, mental age, adaptive functioning, communication, and social skills scores at baseline than nonresponders. Surprisingly, children in the high responder group exhibited lower motor skills scores, more behavior problems, and more autistic symptoms, as reported by their parents. Instead, the authors [44] found no between-group differences in daily living skills and symptom severity when measured through the Autism Screening Questionnaire [66].

Hedvall et al. [35] evaluated the clinical predictors of 2-year treatment outcomes in preschoolers with ASD. They analyzed individual factors that differed significantly between children who gained the most (GM group) and children who lost the most (LM group), after an EIBI treatment, according to the Vineland Composite Score. The authors found that children in the LM group had been referred at significantly lower ages [35]. Furthermore, autistic-type behavior problems, as assessed through the Autistic Behavior Checklist (ABC [67]), were significantly more severe in the LM group as compared to the GM group. In the GM group, the number of children who had passed the expected developmental milestones at the 18-month check-up was significantly higher. Furthermore, the GM group had a significantly higher cognitive level and produced a higher number of words from the beginning. However, only the cognitive level at intake made a unique statistically significant contribution to the prediction model [35].

## 4. Discussion

The current paper systematically reviewed studies on predictors and moderators of response to different types of evidence-based treatment for children with ASD, namely EIBIs and NDBIs. Specifically, our goal was to shed light on the relationships between pre-treatment child characteristics and specific treatment outcomes, covering different aspects of functioning (i.e., social, communicative, adaptive, cognitive, motor, global functioning, play, and symptom severity). Figure 2 summarizes the relationship between predictors and dependent variables (panel a) and the responders’ profile for the two treatment approaches (panel b).

According to our results, it emerges that cognitive functioning is the variable with the greatest predictive power for EIBIs: as shown in Figure 2a, the cognitive level at intake influences post-treatment cognitive functioning, adaptive behavior, communication, the severity of symptoms, and motor skills.

On the other hand, NDBI studies underline the predictive effect of different variables such as communication, play and social skills, specifically social attention, and social avoidance sub-categories.

These results could reflect the idea that some behavioral aspects may act as prerequisites, increasing the effectiveness of specific treatments. In fact, NDBIs are based on incidental teaching procedures conducted through play activities and social interaction, exploiting child intrinsic motivation [5]. Thus, the variables listed above may represent pivotal elements of the treatment itself. Specifically, for NDBIs, social interaction and functional play allow the practitioners to work effectively following the child’s motivation.

On the contrary, since EIBIs are delivered in a more structured setting, they are less bound to intrinsic motivation and consequentially less constrained by individual characteristics. In line with this idea, our results showed that it is not possible to define a specific EIBI responders profile. In fact, the only variable with a meaningful predictive power is cognitive functioning, probably because it could serve as a scaffolding for other acquisitions.

However, the communication and socialization variables seem to be pivotal and cross-cutting elements for both treatments, acting as prerequisites for learning.

Results also revealed a lack of data on the potential predictive power of communication on symptom severity and challenging behavior. It is well known that communication plays a fundamental role in emerging of dangerous problem behaviors [68,69]. However, only one study observed the association between communication and symptom severity, confirming its significant predictive effect [34].

Results also showed that each treatment focused on specific dependent variables. In fact, in EIBI studies, more attention is given to cognitive, social, and adaptive behavior skills, whereas in NDBI studies, the most investigated dependent variables are communication, motor skills, game skills, and global development (i.e., DQ).

This distinction is evident if we look at the play skill variables, which are central in NDBI studies while being ignored in EIBI studies. The authors usually focused on specific aspects of the child’s functioning, which are those most triggered by the treatment itself. This leads to a gap in scientific knowledge since they start from the treatment characteristics neglecting individual variables that could contribute to the treatment effectiveness.

One controversial aspect to highlight is the effect of age on treatment response. In fact, although most studies agree on the importance of early intervention, showing better response in younger children (for a review, see [3], our results suggest that the evidence in favor of this idea is scarce for both treatment approaches. However, it should be noticed that the studies included in our review were conducted on very young children, so it could be that a ceiling effect might cover the effect on treatment outcome and that other factors may have weighted greater in moderating treatment effects in the selected studies. Future studies should clarify the role of age by considering treatment effects in different age groups.

Some limitations suggest caution in interpreting our results. First of all, our search strategy could have been limited by the reliance on a single database for the identification of potentially eligible studies. However, we choose PubMed as it represents one of the most reliable resources for scientific literature. Prospective studies are needed to compare our results with those arising from different databases in order to critically review potential differences in specific variables moderating treatment response. However, we should take into account the scarcity of current studies, preventing a comprehensive overview of the field.

Another important remark concern the exclusion of focused and parental-mediated treatments, possibly shrinking the heterogeneity of treatments observed by our research. Nevertheless, this methodological choice was led by our interest in identifying the effects of treatment on unspecific variables. Indeed, comprehensive treatments provide us with a complete picture of the child’s characteristics that may influence treatment outcomes, given their multidimensional nature. Furthermore, we excluded parental-mediated therapies to reduce the effect of intervening variables related to the treatment fidelity.

## 5. Concluding Remarks and Future Directions for Research

Research on predictors of treatment response is helpful in guiding clinicians in choosing and tailoring the proper treatment for an individual with ASD according to his/her specific learning profile and developmental levels. Although the paucity of the studies reviewed precludes any definitive conclusions, our results provide preliminary evidence of which characteristics an individual should have to optimally benefit from a specific ABA treatment approach. However, although it could be possible to identify some pre-treatment variables that influence treatment outcomes, it is not obvious to define a specific responder’s profile.

The literature examined in this report expands scientific knowledge about prognostic factors associated with specific treatments, showing that if a child is equipped with some skills before treatment, his/her improvements will be greater. However, the question of which treatment is most effective for a given child remains open. In this case, future research should focus on the definition of evidence-based decision-making models able to capture those individual and family factors through which a specific intervention will exert its effects. The decision-making aspect has probably been neglected in favor of a dichotomous approach. The limitation of the latter is that no overlap between models is allowed. Indeed, in the face of the lack of elective profiles for each specific model, each patient could benefit from different cross-cutting aspects of multiple models. In other words, rather than trying to figure out if a certain child would better fit with a specific treatment, research efforts should focus on understanding how to adapt the model to each child’s learning and developmental profile.

In line with this, in a recent study [70] comparing the effects of two types of interventions (i.e., DTT and ESDM), it was found that neither the initial developmental profile nor the severity of the symptoms differentially moderated the outcomes of each treatment. According to the authors, one possible explanation was that, while maintaining a high level of fidelity, clinicians involved have modified the delivery style according to initial child characteristics by using more naturalistic strategies in the DTT condition and more structured procedures in the ESDM condition, depending on the severity of the child’s functioning [70]. Another possible explanation might be that ABA-based treatments share key practices and principles. Therefore, rather than choosing the intervention according to the “brand name” of each treatment, a more personalized approach is needed so as to retrieve the best fit between child characteristics and evidence-based “ingredients” of each treatment [17,23].

## Figures and Tables

**Figure 1 behavsci-12-00267-f001:**
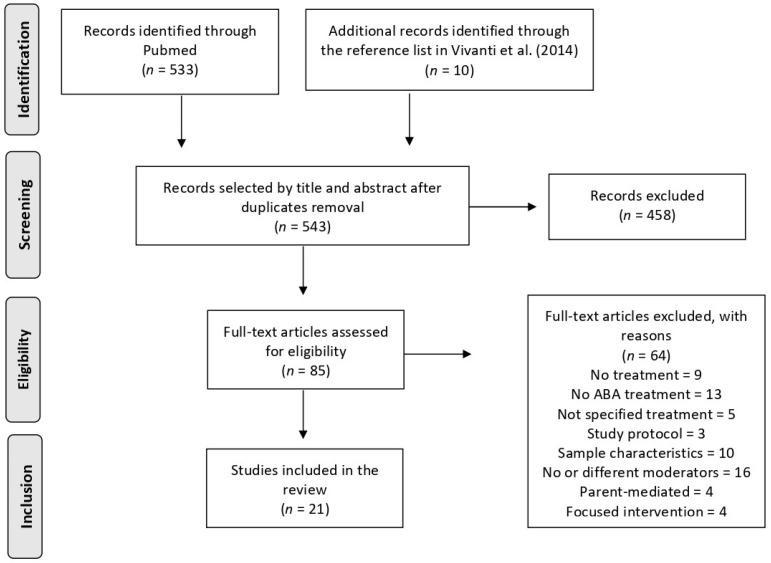
Adapted from the PRISMA flow diagram [24], showing the steps of the selection process undergone by the studies in this review.

**Figure 2 behavsci-12-00267-f002:**
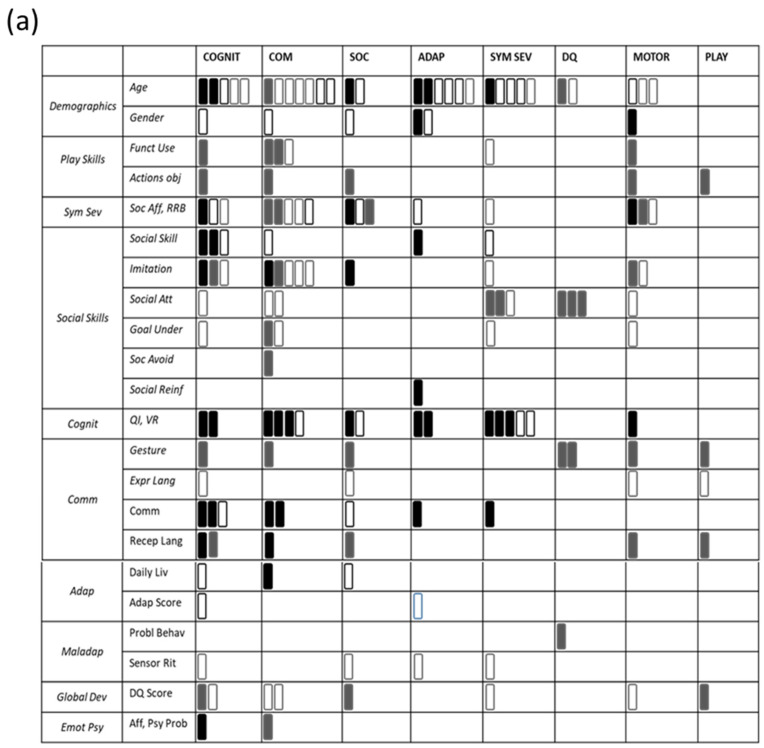
Panel (**a**) shows the effects between predictors (in column) and dependent variables (in line). Full rectangles (black for EIBI studies, grey for NDBI studies) indicate that a significant predictive effect has been found for that dependent variable. Empty rectangles (with black outlines for EIBI studies and grey outlines for NDBI studies) indicate that the effect was not significant for that dependent variable. Panel (**b**) shows responders’ profile for each treatment approach, EIBI on the top and NDBI at the bottom. Solid lines: Hedval et al. [35]; Rhombus texture: Remington et al. [44]; Dotted texture: Fossum et al. [31]; Horizontal lines: Laister et al. [36]; Vertical lines: Sherer and Schreibman [37]; Diagonal lines: Sinai-Gavrilov et al. [27].

**Table 1 behavsci-12-00267-t001:** Summary of Studies Included in Review (*n* = 21).

Study	Study Design and Participants	Treatment Features	Dependent Variables	Moderators	Main Findings
Vivanti et al., 2013 [25]	Interventional single group Total sample: *n* = 21 *M*age = 38 ± 11.5 months	Group-based ESDM Intensity: 15–25 h/week Duration: 1 year	Visual Reception; Fine Motor; Receptive Language; Expressive Language; Symptoms severity	Functional use of objects; Imitation; Goal Understanding; Symptom severity; Social Attention; Chronological age; DQ	Functional use of objects was positively associated with Visual Reception, Fine Motor and Expressive Language domains gains and alone explained 70% of the variance of the Visual Reception domains; Imitation was positively associated with Visual Reception, Fine Motor and Receptive Language domains gains; Goal Understanding was positively associated with Receptive Language gains and explained alone 30% of the variance of Receptive Language; Symptom severity was negatively associated with Receptive and Expressive Language gains and alone explained 40% of the variance of Expressive Language; Chronological age was negatively associated with Expressive Language; Functional use of objects was not associated with Receptive Language, Symptoms severity Social attention was not associated with Visual Reception, Fine Motor, Receptive and Expressive Language, Symptoms severity; Goal Understanding was not associated with Visual Reception, Fine Motor, Expressive Language, Symptoms severity; Imitation was not associated with Expressive Language, Symptoms severity; Chronological Age was not associated with Visual Reception, Fine Motor, Receptive Language, Symptoms severity; DQ was not associated with Visual Reception, Fine Motor, Receptive and Expressive Language, Symptoms severity; Symptom severity was not associated with Visual Reception, Fine Motor, Symptoms severity
Vivanti et al., 2016 [26]	Observational case control study Total sample: *n* = 60 Younger group (18–48 months): *n* = 32 *M*age = 33.25 ± 7.2 years Older group (48–62 months): *n* = 28 *M*age = 49.54 ± 5.36 years	Group-based ESDM Intensity: 15–25 h/week	Verbal DQ; NonVerbal DQ; Adaptive behavior; Symptom severity	Chronological age	Chronological age was inversely associated with Verbal DQ gains; Changes in NonVerbal DQ, Adaptive behavior, Symptom severity were not associated with Chronological age
Sinai-Gavrilov et al., 2020 [27] *	Repeated measures factorial design Total sample: *n* = 51 PB-ESDM: *n* = 26 *M*age = 43.65 ± 7.37 years MDI group: *n* = 25 *M*age = 45.12 ± 4.8 years	preschool-based ESDM Intensity: 44 h/week Duration: 8 weeks	DQ; Communication; Daily living skills; Socialization; Motor skills; Adaptive behavior	Symptom severity; Fine Motor Skills; Visual Reception; Receptive Language; Expressive Language; DQ; Adaptive behavior; Chronological Age; Gender	High responders had lower Symptom severity, higher DQ and higher Adaptive behavior compared to low responders; No between-group changes emerged for Gender and Chronological age
Latrèche et al., 2021 [28]	Longitudinal cohort study Total sample: *n* = 95 *M*age = 2.81 ± 0.65 years ESDM-AF+ group: *n* = 25 *M*age = 2.83 ± 0.48 years ESDM-AF- group: *n* = 26 *M*age = 2.68 ± 0.68 years CT-AF+ group: *n* = 16 *M*age = 3.04 ± 0.55 years CT-AF- group: *n* = 14 *M*age = 2.62 ± 0.67 years	ESDM Intensity: 20 h/week	DQ	Attention to face	Children with more Attention to face at baseline (ASD-AF + ) demonstrated statistically significantly higher DQ scores over time, especially in the verbal domain
Robain et al., 2020 [29]	Observational case-control study Total sample: *n* = 60 *M*age = 3.0 ± 0.8 years ESDM-GR group: *n* = 9 *M*age = 2.61 ± 0.39 years ESDM-SR group: *n* = 13 *M*age = 2.57 ± 0.37 years CT-GR group: *n* = 23 *M*age = 3.36 ± 0.72 years CT-SM group: *n* = 15 *M*age = 3.20 ± 1.10 years	ESDM Duration: 1 year	RRB; Social Affect; Symptom severity; DQ	Social orienting; Maladaptive behavior	Social orienting predicted changes in Social Affect, Symptom severity and DQ changes after ESDM treatment, in that ESDM-SR group reported lower Symptom severity and higher DQ over time; Higher Maladaptive behavior were associated with lower DQ after treatment but not with DQ changes over time Social orienting was not predictive of RRB mean change over time
Contaldo et al., 2020 [30]	Pre-post single-group design Total sample: *n* = 32 *M*age = 28.8 ± 6.5 months	Community-based ESDM Intensity: 4 h/week Duration: 1 year	Communication; Socialization; Cognition and Play; Motor skills; Number of learning objective acquired by each child in a month; Adaptive functioning; First Communicative Gestures; Actions with objects; Imitation; Receptive Lexical Quotient; DQ	First Communicative Gestures; Actions with objects; Receptive Lexical Quotient; Word Production; Imitation; DQ; Symptom severity; Chronological age	Number of First Communicative Gestures were positively associated with gains in Communication, Socialization, Cognition and Play, and Motor skills and with the number of learning objective acquired by each child in a month; Actions with objects were positively associated with gains in Socialization, Cognition and Play, and Motor skills and with the number of learning objective acquired by each child in a month; Receptive Lexical Quotient was positively associated with gains in Socialization, Cognition and Play, and Motor skills and with the number of learning objective acquired by each child in a month; DQ was positively associated with gains in Socialization, Cognition and Play, and with the number of learning objective acquired by each child in a month Symptom severity was negatively associated with gains in Socialization and Motor skills and with the number of learning objective acquired by each child in a month; Actions with objects and Receptive Lexical Quotient were not associated with the Communication domain; Symptom severity was not associated with Communication and Cognition and Play domains; DQ was not associated with Communication and Motor skills domains; Chronological age, Word Production, and Imitation were not associated with any changes neither with the rate of learning objectives acquisition
Fossum et al., 2018 [31] *	Prospective single-subject design Total sample: *n* = 57 *M*age = 47.84 ± 8.86 years	PRT; Duration: 1 year	Communication	Toy contact; Social avoidance; Verbal self-stimulatory behavior; Positive affect; Cognitive abilities; Expressive language; Chronological age; Symptoms severity	Higher levels of Expressive Language, Cognitive ability, Toy contact, Positive affect, lower Social avoidance and Verbal self-stimulatory behavior at baseline appeared to predict treatment outcome in Expressive language; Chronological age and Symptom severity were not predictive of Communication outcome
Schreibman et al., 2009 [32] *	Single-subject multiple baseline design across participants Total sample: *n* = 6 *R*age = 2–4 years Nonresponders with high toy contact: *n* = 3 *M*age = 26 month Nonresponders with low avoidance: *n* = 3 *M*age = 34.67 months	PRT + DTT Intensity: 18 h/week	Communication	Toy contact; low avoidance	Higher Toy contact is a key characteristic of PRT treatment responders, whereas low avoidance is not associated with a better response to PRT
Tiura et al., 2017 [33]	Longitudinal study Total sample: *n* = 35 *M*age = 2.10 years	EIBI Intensity: 9–30 h/week	Communication; Social–emotional; Adaptive behavior; Physical development	Cognitive functioning; Speaking English as the primary language; Gender; Symptom severity; Chronological age	Children with higher Cognitive functioning predicted rapid growth across the four dependent variables; Participants who spoke English as a primary language had faster growth rates in the social–emotional and physical development domains; Male participants tended to improve more quickly in the areas of Adaptive behavior and Physical development; Children with higher Symptom severity tended to improve at a slower pace in the Physical development domain; Speaking English as the primary language was not associated with Adaptive behavior and Communication; Gender was not associated with Social–emotional and Communication; Symptom severity was not associated with Adaptive, Social–emotional and Communication; Chronological age did not predict growth rate
Magiati et al., 2011 [34]	Longitudinal study Total sample: *n* = 36 *M*age = 3.4 ± 0.6 years	EIBI Intensity: 15–40 h/week; Duration: 22–90 months	Cognitive functioning; Language skills; Adaptive behavior; Symptom severity	Cognitive functioning; Language skills; Adaptive behavior; Symptom severity; Chronological age	Cognitive functioning, Language and Adaptive behavior skills were predictive of outcomes after 4–5 years of intervention; Chronological age was not associated with treatment outcomes; Symptom severity was not predictive of outcomes
Hedvall et al., 2015 [35]	Observational Case Crossover Initial group: *n* = 198 *M*age 39 ± 8.5 months GM group: *n* = 30 *M*age = 41 ± 8.7 months LM group: *n* = 23 *M*age = 33 ± 7.9 months	EIBI Intensity: 15–40 h/week Duration: 2 years	Adaptive behavior	Cognitive functioning; Chronological age; Developmental milestones at 18 months; Regression; Symptom severity; Gender	Children in the LM group had been referred at significantly lower ages; Symptom severity was significantly higher in the LM group as compared to the GM group; Most of the children in the GM group significantly had passed the expected developmental milestones at the 18-month check-up; The GM group had a significantly higher cognitive level
Laister et al., 2021 [36] *	Longitudinal pre-post design Total sample: *n* = 56 *M*age = 41.96 ± 7.5 years	ESDM Duration: 12 months	Social skills; Language skills	Verbal DQ; Nonverbal DQ; Fine Motor skills; Gross Motor skills; Visual Reception; Receptive Language; Expressive Language; Gestural approach behavior; Social approach behaviors; Problem behavior; Chronological Age	Verbal and Nonverbal DQ, Fine Motor skills, Visual Reception, and Gestural Approach Behavior were significantly higher for the High responder group who gains the most in social and language skills; Receptive Language, Expressive Language, Problem Behavior, Chronological Age, and Social approach behaviors were not significantly different for the two groups; Gestural approach behaviors were found to significantly predict Verbal and Nonverbal DQ;
Sherer and Schreibman, 2005 [37] *	Multiple baseline design across participants Total sample: *n* = 6 Responders: *n* = 3 *M*age = 3:3 years; Nonresponders: *n* = 3 *M*age = 4:2 years	PRT Intensity: 10 h/week	Language Skills; Adaptive Behavior; IQ; Symptom severity; Play skills	Toy Contact/Object Manipulation; Approach behaviors; Avoidant behaviors; Verbal Self-Stimulatory behaviors; Nonverbal Self-Stimulatory Behaviors	Children with higher Toy contact/Object Manipulation, Approach behavior and Verbal Self-Stimulatory behaviors but lower Avoidant behaviors responded better to treatment than children in the nonresponder group
Smith et al., 2010 [38]	Observational case control study Total sample: *n* = 45 *M*age = 50 ± 10 months	EIBI Intensity: 14 h/week Duration: 12 months	Expressive Language; Receptive Language Adaptive behavior; Symptoms severity; Problem behavior	IQ	Significant main effects of IQ on Expressive and Receptive Language, Communication, Adaptive behavior and Symptom severity were found; Non-significant effects of IQ on Problem Behavior emerged
Sallows and Graupner, 2005 [39]	Interventional Parallel Group Study Total sample: *n* = 23 UCLA group: *n* = 13 *M*age = 33.2 ± 3.89 months Parent group: *n* = 10 *M*age = 34.2 ± 5.06 months	UCLA EIBI Intensity: 40 h/week Duration: 2 years	IQ; Language skills; Social skills; Early Learning Measure	IQ; Daily Living skills; Social skills; Communication; Symptoms severity; Early Learning Measure	IQ was best predicted by Early Learning Measure, IQ, Social skills, lower Symptom severity; Daily Living skills and Communication did not predict changes in IQ; Language skills was predicted by the ability to imitate, Daily Living skills and Communication; IQ, Social skills and Symptom severity did not predict changes in Language skills; Social skills was predicted by the ability to imitate, Early Learning Measure, Symptom severity; Daily Living skills, IQ and Communication did not predict Social skills
Smith et al., 2015 [40]	Interventional Single Group Study Total sample: *n* = 71 Mage = 3.27 ± 0.65 years	EIBI Intensity: ≥15 h/week Duration: 24 months	IQ; Adaptive behavior; Symptom severity; Social skills; Communication; RRB	Chronological age; IQ; Social Engagement; Social approach; Joint attention; Imitation; Sensorimotor rituals	Higher values of outcome variables at intake predicted better outcome in IQ, Adaptive behavior, Symptom severity; Lower Chronological age predicted better outcome for IQ, Adaptive behavior, and Communication; Higher IQ predicted IQ level, Adaptive behavior and lower Symptom severity; Higher Social Engagement scores at intake predicted higher IQ and Adaptive behavior; Sensorimotor rituals did not predict any outcome; Social engagement did not predict Symptom severity outcome; Chronological age did not predict Social Interaction deficit and Symptom severity; IQ did not predict Social Interaction and RRB
Klintwall and Eikeseth, 2012 [41]	Interventional Single Group Study Total sample: *n* = 21 *M*age = 3.7 *R*ge = 2.3–4.11 years	EIBI Intensity: 20 h/week	Learning rate of Adaptive behavior	Automatic reinforcers; Socially-mediated reinforcers; Chronological age; Adaptive behavior	The number of Socially-mediated reinforcers was found to be a significant predictor of increase in the learning rate, vice-versa for the number of automatic reinforcers; Chronological age was also found to be a significant predictor of learning rate: older children exhibited larger treatment gains; Adaptive behavior did not predict learning rate
Pellecchia et al., 2016 [42]	Interventional Single group Study Total sample: *n* = 152 *M*age = 6 ± 0.9 years	STAR Duration: 3 years	IQ	Language and Communication; Adaptive behavior; Challenging behavior; Symptom severity; Social skills; Chronological age; Symptoms associated with co-occurring psychiatric difficulties	Social anxiety symptoms significantly predicted IQ outcome, in that increased social phobia was associated with a decrease in cognitive functioning; Chronological age significantly predicted IQ changes, in that lower age was associated with a decrease in cognitive functioning;Language and Communication, Adaptive skills, Challenging behaviors, Symptom severity, Social skills and other co-occuring psychiatric difficulties did not predict any change in IQ
Ben-Itzchak et al., 2014 [43]	Interventional Parallel Study Total sample: *n* = 46 *M*age = 25.5 ± 3.95 months	ABA-based treatment Intensity: 20 h/week	Communication; Daily Living skills; Socialization; Motor skills; Fine Motor; Visual Reception; Receptive Language; Expressive Language; Symptoms severity	IQ	Higher IQ was associated with increases in Communication, Daily Living skills, Socialization; A lower IQ was associated with an increase in Fine Motor and Receptive Language; IQ did not predict Symptom severity
Remington et al., 2007 [44] *	Interventional Parallel Group Study Total sample: *n* = 44 *R*ange = 30–40 months EIBI group *n* = 23 *M*age = 35.7 ± 4.0 months Control group *n* = 21 *M*age = 38.4 ± 4.4 months	EIBI Intensity: 25.6 h/week Duration: 2 years	IQ; Language skills	IQ; Adaptive behavior; Communication; Socialization; Daily Living skills; Motor skills; Problems behavior; Symptoms severity	Children who responded better to intervention had higher IQ, higher Adaptive behavior, Communication and Social Skills scores, lower Motor skills scores, more Problems behaviors and higher Symptoms severity; No between-group differences were found for Daily living skills
Eldevik et al., 2012 [45]	Interventional Parallel Group Study Total sample: *n* = 43 EIBI group: *n* = 31 *M*age = 42.2 ± 9 months TAU group: *n* = 12 *M*age = 46.2 ± 12.4 months	EIBI Intensity: 10–20 h/week Duration: 2 years	Communication; Socialization; Daily Living skills; Adaptive behavior; IQ	Chronological age; IQ; Adaptive behavior; Gender; Diagnosis	Chronological age positively correlated with gains in Adaptive behavior scores; IQ positively correlated with changes in the Socialization domain;Gender was not associated with any of the dependent variables

Notes. * indicates responders’ profile studies. CT: community treatment; CT-AF-: CT subgroups with lower attention to face; CT-AF+: CT subgroups with higher attention to face; CT-GR: CT geometrical responder subgroups; CT-SR: CT social responder subgroups; DQ: Developmental Quotient; ESDM: Early Start Denver Model; EIBI: Early Intensive Behavioral Intervention; ESDM-AF-: ESDM subgroups with lower attention to face; ESDM-AF+: ESDM subgroups with higher attention to face; ESDM-GR: ESDM geometrical responder subgroups; ESDM-SR: ESDM social responder subgroups; GM: children who gained the most; LM: children who lost the most; MDI: Multidisciplinary Developmental Intervention; PB-ESDM: Preschool-based ESDM; RRB: Restricted interest and Repetitive Behaviors; STAR: Strategies for Teaching based on Autism Research [46]; TAU: Treatment As Usual.

## Data Availability

Not applicable.

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
