# Peer review of "Examining Predictors of Different ABA Treatments: A Systematic Review"

_behavsci, 2022, doi:10.3390/bs12080267_

Round 1

Reviewer 1 Report

Thank you for providing me with this opportunity to review your work. 

1. Line 87: Authors need to include the initials of two reviewers that carried out the literature review. Additionally, were there any other databases utilized for systematic search besides PubMed?

2. Line 110: Earlier review by Vivanti et al. was utilized to screen for studies. This needs to be mentioned earlier in the methods section.

3. In the PRISMA flow diagram, the databases that were used for the search need to be mentioned. For example, PubMed.

4.  In the PRISMA flow diagram, the other sources that were used for the search need to be mentioned. 

Author Response

Line 87: Authors need to include the initials of two reviewers that carried out the literature review.

Response: Done as suggested.

Additionally, were there any other databases utilized for systematic search besides PubMed?

Response: As mentioned in the method section, we have used only one database search, i.e., Pubmed, and we have highlighted it in the discussion part, as a limitation.

Line 110: Earlier review by Vivanti et al. was utilized to screen for studies. This needs to be mentioned earlier in the methods section.

Response: Done as suggested.

  1. In the PRISMA flow diagram, the databases that were used for the search need to be mentioned. For example, PubMed.

Response: Done as suggested.

  1. In the PRISMA flow diagram, the other sources that were used for the search need to be mentioned.

Response: Done as suggested.

Reviewer 2 Report

Dear authors,

The article presented is of high quality and has been carried out according to the desirable criteria of a research article and, of course, the steps of a review and the PRISMA recommendations have been followed. However, I put forward some suggestions for improvement:

1. INTRODUCTION:

- I recommend including in the first paragraph of the introduction a more current citation of the DSM-V, since a new version (DSM-V-TR) came out in early 2022:

American Psychiatry Association (2022). Diagnostic and Statistical Manual of Mental Disorders (DSM-V-TR), 5th ed. American Psychiatry Association.

- It is important to cite this manual according to MDPI standards.

- The introduction, although brief, clearly summarizes the main characteristics of people with ASD and highlights treatments based on ABA.

- The research cited is not current. It is recommended to include previous studies from the last 5 years.

- The research objective is to conduct a systematic review of the literature on characteristics of children that may influence response to specific treatment approaches.

2. METHOD:

- Inclusion and exclusion criteria have been well defined from the beginning of the article.

- The flow chart (Figure 1) shows the number of articles reviewed. It is well understood. Despite the large number of articles found in the initial searches, only 21 that meet the criteria specified by the authors are analyzed. I consider it important to include in the wording before Figure 1 that the total number of articles reviewed was 21. Reading the description on lines 113 to 122, it appears that 26 articles were reviewed instead of 21.

3.       RESULTS:

- Table 1 perfectly captures the information for the 21 articles. Lately, this type of table is being used to analyze articles and it fits very well with the content explained.

- It is surprising that most of the studies are not current, which justifies the need for such an article. Authors are encouraged to incorporate this in the introduction even though it is covered in lines 474-475.

- The results are accurately analyzed and the reviewed articles are cited from Table 1. Each developmental domain (communication, adaptive behavior, etc., ...) is rigorously analyzed in subsections (3.1... 3.11).

- Figure 2 exemplifies and details what was read previously, summarizes the relationship between predictors and dependent variables and the profile of responders for the two treatment approaches.

4. DISCUSSION:

- The results discussed in the previous section are adequately discussed.

- The authors reflect that a limitation lies in using only the PubMed database. It is suggested to include as prospective research to extend the search to more relevant databases in the current scientific panorama and to check if these results found differ from those found in future searches. For example, the variable age in the response to treatment.

- It would also be important to include as a limitation, the scarcity of current studies focused on this subject, since most of them do not belong to the last five years.

5. CONCLUDING REMARKS AND FUTURE DIRECTIONS FOR RESEARCH:

- We reflect on the predictors of response to treatment and close the article with expanded information on the phenomenon analyzed. The research prospects included are adequate.

6. REFERENCES

- It is recommended to expand the references with more current articles.

- It is recommended to include the url of the complete doi, instead of doi:

       For example: https://doi.org/10.1007/s10803-020-04430-6  instead of: doi: 10.1007/s10803-020-04430-6.

- Overall, this section is well done.

The article presents a good format and analysis of the literature, but I strongly recommend improving the suggested aspects.

Author Response

  1. INTRODUCTION:

 - I recommend including in the first paragraph of the introduction a more current citation of the DSM-V, since a new version (DSM-V-TR) came out in early 2022:

American Psychiatry Association (2022). Diagnostic and Statistical Manual of Mental Disorders (DSM-V-TR), 5th ed. American Psychiatry Association.

- It is important to cite this manual according to MDPI standards.

Response: Done as suggested.

 - The introduction, although brief, clearly summarizes the main characteristics of people with ASD and highlights treatments based on ABA.

- The research cited is not current. It is recommended to include previous studies from the last 5 years.

Response: Here, the Reviewer highlighted an important gap in autism scientific literature related to the paucity of current papers on treatment variability issues. However, we have included an interesting article recently written by Chen and co-workers, suggesting rethinking autism intervention science (lines 77-79).

- The research objective is to conduct a systematic review of the literature on characteristics of children that may influence response to specific treatment approaches.

 METHOD:

 - Inclusion and exclusion criteria have been well defined from the beginning of the article.

- The flow chart (Figure 1) shows the number of articles reviewed. It is well understood. Despite the large number of articles found in the initial searches, only 21 that meet the criteria specified by the authors are analyzed. I consider it important to include in the wording before Figure 1 that the total number of articles reviewed was 21. Reading the description on lines 113 to 122, it appears that 26 articles were reviewed instead of 21.

Response: We thank the Reviewer for this important remark and we have corrected the number of articles accordingly.

 RESULTS:

 - Table 1 perfectly captures the information for the 21 articles. Lately, this type of table is being used to analyze articles and it fits very well with the content explained.

- It is surprising that most of the studies are not current, which justifies the need for such an article. Authors are encouraged to incorporate this in the introduction even though it is covered in lines 474-475.

Response: Done as suggested. Please, see line 74.

 - The results are accurately analyzed and the reviewed articles are cited from Table 1. Each developmental domain (communication, adaptive behavior, etc., ...) is rigorously analyzed in subsections (3.1... 3.11).

- Figure 2 exemplifies and details what was read previously, summarizes the relationship between predictors and dependent variables and the profile of responders for the two treatment approaches.

 DISCUSSION:

 - The results discussed in the previous section are adequately discussed.

- The authors reflect that a limitation lies in using only the PubMed database. It is suggested to include as prospective research to extend the search to more relevant databases in the current scientific panorama and to check if these results found differ from those found in future searches. For example, the variable age in the response to treatment.

Response: According to the Reviewer, we have added this suggestion in lines 467-469.

 - It would also be important to include as a limitation, the scarcity of current studies focused on this subject, since most of them do not belong to the last five years.

Response: Done as suggested.

 CONCLUDING REMARKS AND FUTURE DIRECTIONS FOR RESEARCH:

 - We reflect on the predictors of response to treatment and close the article with expanded information on the phenomenon analyzed. The research prospects included are adequate.

 REFERENCES

 - It is recommended to expand the references with more current articles.

Response: We agree with the Reviewer that the lack of current research in the field prevent from a comprehensive overview of the issue. However, thanks to his suggestions, we have now pointed out this remark in different part of our paper. Also, we have now added a very recent paper by Chen et al. (2022).

 - It is recommended to include the url of the complete doi, instead of doi:

        For example: https://doi.org/10.1007/s10803-020-04430-6  instead of: doi: 10.1007/s10803-020-04430-6.

Response: Done as suggested.

 - Overall, this section is well done.

The article presents a good format and analysis of the literature, but I strongly recommend improving the suggested aspects.
